# The Autophagy Protein Pacer Positively Regulates the Therapeutic Potential of Mesenchymal Stem Cells in a Mouse Model of DSS-Induced Colitis

**DOI:** 10.3390/cells11091503

**Published:** 2022-04-30

**Authors:** Cristian A. Bergmann, Sebastian Beltran, Ana Maria Vega-Letter, Paola Murgas, Maria Fernanda Hernandez, Laura Gomez, Luis Labrador, Bastián I. Cortés, Cristian Poblete, Cristobal Quijada, Flavio Carrion, Ute Woehlbier, Patricio A. Manque

**Affiliations:** 1Center for Integrative Biology (CIB), Faculty of Science, Universidad Mayor, Santiago 7500000, Chile; cbergmannmunoz@gmail.com (C.A.B.); sebastian.beltran.v@gmail.com (S.B.); paolamurgas@gmail.com (P.M.); fernandahernandezberrios@gmail.com (M.F.H.); laura.gomezg1995@gmail.com (L.G.); alexardy@gmail.com (L.L.); cortes.bastian@gmail.com (B.I.C.); 2Escuela de Tecnología Médica, Universidad Mayor, Santiago 7500000, Chile; 3Laboratorio de Inmunología Celular y Molecular, Centro de Investigación Biomédica, Facultad de Medicina, Universidad de Los Andes, Santiago 7620001, Chile; avegaletter@gmail.com; 4Centro de Investigación e Innovación Biomédica, Universidad de los Andes, Santiago 7620157, Chile; 5Escuela de Biotecnología, Facultad de Ciencias, Universidad Mayor, Santiago 7500000, Chile; 6Laboratorio de Morfofisiopatología y Citodiagnóstico, Escuela de Tecnología Médica, Facultad de Ciencias, Universidad Mayor, Santiago 7500000, Chile; cristian.poblete@umayor.cl; 7Servicio de Anatomía Patológica, Hospital Clínico de la Universidad de Chile, Santiago 8380456, Chile; cristobal.quijadacruz@gmail.com; 8Programa de Inmunología Translacional, Facultad de Medicina, Universidad del Desarrollo Clínica Alemana, Santiago 7590943, Chile; flaviocarrion@yahoo.com; 9Departamento de Investigación, Postgrado y Educación Contínua (DIPEC), Facultad de Ciencias de la Salud, Universidad del Alba, Santiago 8320000, Chile; 10Center for Genomics and Bioinformatics (CGB), Faculty of Science, Universidad Mayor, Santiago 7500000, Chile; 11Centro de Oncologia de Precision (COP), Escuela de Medicina, Universidad Mayor, Santiago 7500000, Chile

**Keywords:** autophagy, PACER, KIAA0226L, RUBCNL, therapy, inflammatory bowel disease, colitis, mesenchymal stem cells

## Abstract

Mesenchymal stem cells (MSC) have emerged as a promising tool to treat inflammatory diseases, such as inflammatory bowel disease (IBD), due to their immunoregulatory properties. Frequently, IBD is modeled in mice by using dextran sulfate sodium (DSS)-induced colitis. Recently, the modulation of autophagy in MSC has been suggested as a novel strategy to improve MSC-based immunotherapy. Hence, we investigated a possible role of Pacer, a novel autophagy enhancer, in regulating the immunosuppressive function of MSC in the context of DSS-induced colitis. We found that Pacer is upregulated upon stimulation with the pro-inflammatory cytokine TNFα, the main cytokine released in the inflammatory environment of IBD. By modulating Pacer expression in MSC, we found that Pacer plays an important role in regulating the autophagy pathway in this cell type in response to TNFα stimulation, as well as in regulating the immunosuppressive ability of MSC toward T-cell proliferation. Furthermore, increased expression of Pacer in MSC enhanced their ability to ameliorate the symptoms of DSS-induced colitis in mice. Our results support previous findings that autophagy regulates the therapeutic potential of MSC and suggest that the augmentation of autophagic capacity in MSC by increasing Pacer levels may have therapeutic implications for IBD.

## 1. Introduction

Mesenchymal stem cells (MSC) display high self-renewing properties and can differentiate into a variety of cell types, including osteoblasts, chondrocytes and adipocytes, maintaining adult mesenchymal tissues [1,2,3]. MSC are adult progenitor cells and are found in almost all postnatal tissues and organs with similar morphological and functional features, which include the capacity for multi-lineage differentiation as well as immunosuppressive and pro-angiogenic characteristics [4,5,6,7]. Stem cell therapy with MSC shows promise for an increasing number of autoimmune, degenerative and inflammatory diseases, including inflammatory bowel disease (IBDs), graft versus host disease (GvHD), systemic lupus erythematosus (SLE) and rheumatoid arthritis [8,9]. IBD mouse models using dextran sulfate sodium (DSS) to induce colitis are extensively used to study the immunomodulatory properties of MSC [10,11]. DSS-induced colitis mouse models mimic some key immunological and histopathological features of IBDs in humans, hence these models can thus be valuable tools to test the evolving therapeutic strategies in a preclinical setting [12].

Tumor necrosis factor alpha (TNFα) is a pleiotropic cytokine involved in a wide range of pathological processes, including IBD. Increased levels of TNFα have been reported in serum, stool or mucosal biopsy specimens of IBD patients [13,14,15,16]. Furthermore, the successful treatment of IBD in patients with TNFα chimeric monoclonal antibodies (cA2 or infliximab) established a clear association of the role of this cytokine in the pathogenesis of IBD [17]. However, many patients with IBD show primary non-response to this therapy or demonstrate loss of response over time (secondary non-response) [18]. Hence, stem cells have emerged as an attractive novel approach for clinical therapy for patients with IBD [19]. The administration of MSC in both mice and humans has been shown to modulate the function of all immune cells affecting both innate and adaptive immune responses [20]. MSC can secrete several anti-inflammatory molecules, such as prostaglandin E2 (PGE2), nitric oxide (NO), transforming growth factor 1 beta (TGF1β), hepatocyte growth factor (HGF), interleukin-6 (IL6) and interleukin-10 (IL10), to inhibit the proliferation and function of immune cells, including dendritic cells, T and B lymphocytes, thereby dampening the severity of inflammation. The exposure to an inflammatory tissue environment can modulate the immunosuppressive function of MSC [21]. Thus, TNFα and other pro-inflammatory cytokines, such as interferon ɣ (INFɣ) and interleukin 1 (IL1), are present in the inflammatory tissues and can potentiate the immunosuppressive function of MSC [22,23,24].

Although a consensus exists about the regulatory function of autophagy in MSC toward their stemness, differentiation capacity and immunosuppressive function, the specific effects of modulating autophagy in MSC have been reported in several studies with opposing findings (reviewed in Ref [25]). Autophagy is a highly conserved cellular process that maintains cellular homeostasis by recycling cytoplasmic materials, such as damaged organelles or misfolded proteins, by delivering them to the lysosome for degradation [26]. Through its recycling function, autophagy also provides energy for cellular renovation and homeostasis [27]. Cellular autophagic capacity can be rapidly increased in response to stress signals, such as starvation, hypoxia, infection and inflammation, primarily as a survival mechanism [28]. In MSC, most studies on autophagy have focused on their roles in differentiation [29,30]; whether autophagy plays a role in the immunosuppressive function of MSC remains unknown. Gao et al. demonstrated that the autophagy inhibitor 3-methyladenine (3-MA) weakens the immunosuppressive function of MSC, whereas autophagy inducer rapamycin enhances this function mediated by TGF1β secretion [31]. In this line, MSC treated with rapamycin aggregate facilitating full-layer cutaneous wound healing and regeneration mediated by vascular endothelial growth factor (VEGF) secretion [32]. On the other hand, MSC depleted of Beclin1, a core subunit of distinct phosphatidylinositol 3 kinases (PI3K) complexes, which mediate multiple steps of the autophagy process, lose their regenerative capacity [32]. Hence, the activation of autophagy may be directly involved in the regulation of the immunosuppressive function of MSC. Recently, we and others described a novel autophagy-related protein called Pacer (protein associated with UVRAG as autophagy enhancer) [33,34,35]. Pacer associates with Beclin1 and positively regulates autophagosome maturation by complex association with UV radiation resistance-associated gene protein (UVRAG) and stimulation of Vps34 kinase activity [33,34,36]. Pacer was shown to be modulated through phosphorylation by the mammalian target of rapamycin complex 1 (mTORC1), which prevents the interaction of Pacer with STX17 and the HOPS complex, resulting in disrupted autophagosome maturation [35]. Dephosphorylation of Pacer promotes its acetylation in the RH domain, which was shown to enhance the interaction with the above-mentioned proteins and promote autophagosome maturation [35]. It has been suggested that Pacer has an important role in autophagy-mediated hepatic lipolysis to alleviate liver inflammation and injury [35]. Furthermore, Pacer has been shown to be involved in amyotrophic lateral sclerosis (ALS) pathogenesis where its loss of function led to increased ALS-associated protein aggregation and neuronal cell death [33]. A role for Pacer in MSC autophagic capacity or immunosuppressive function has not been investigated previously. 

Here, we investigated a possible role of Pacer in regulating the immunosuppressive capacity of MSC through its function in the autophagy pathway. We found Pacer to be upregulated upon stimulation with the pro-inflammatory cytokine TNFα. By modulating Pacer expression in MSC, we found that Pacer plays an important role in regulating the autophagy pathway in MSC in response to TNFα stimulation, as well as in regulating the immunosuppressive capacity of MSC toward T-cell proliferation. Furthermore, increased expression of Pacer in MSC enhanced the ability of MSC to ameliorate the symptoms of DSS-induced colitis in mice. Our results suggest that the augmentation of autophagic capacity in MSC by increasing Pacer levels may have therapeutic implications for IBD. Finally, our findings also provide one of the first insights as to how autophagy could modulate the immunoregulatory function of MSC.

## 2. Materials and Methods

### 2.1. Reagents and Mice

Recombinant human TNFα was purchased from Sigma (St. Louis, MO, USA, 0609AFC25). The antibodies used were mouse anti-human Pacer (Novus, B01P, NBH00080183-B01P), anti-rabbit GAPDH (Cell Signaling Technology, Danvers, MA, USA, 2118s), anti-mouse PTGS2 (Cox2) (Beckton Dickinson, Franklin Lakes, NJ, USA, 610203) and custom anti-mouse Pacer manufactured by Abmart raised against a 14 aa peptide of the N-terminal domain of mouse Pacer [33]. Concanavalin A (ConA) (Sigma, SLBR2953) was purchased from Sigma (St. Louis, MO, USA, SLBR2953). SYTOX™ Green dyekit was purchased from Life Technologies (Darmstadt, Germany, S7020), and the antibody used for flow cytometry was APC anti-mouse CD3 (BioLegend, San Diego, CA, USA, 100236). Colitis-grade dextran sulfate sodium (DSS) (MP Biomedicals, Santa Ana, CA, USA, 160110) and C57BL/6 mice were acquired from the Jackson Laboratory and maintained under standard conditions in the animal facility of Universidad Mayor Faculty of Science. All animal procedures were approved by the Animal Welfare and Ethics Committee of Universidad Mayor (Protocol Number, 06/2016-13-2017(E1)). 

### 2.2. Cells

Murine bone marrow MSC were obtained from Gibco (S1502-100). MSC were cultured at 37 °C with 5% CO_2_ in complete alpha modified Eagle’s medium (αMEM; Gibco, Auckland, New Zealand, 12571-063) containing 10% heat-inactivated fetal bovine serum (FBS; Gibco, 10437028, 100 U/mL penicillin and 100 μg/mL streptomycin (Pen-strep, Gibco, DW101203-031-1B). Cells were used between passages 9 and 12. The Human Embryonic Kidney 293 (HEK293T) cell line (Sigma, 12022001) was used for lentiviral particle production using Lenti-ORF clones kit (Origene, Rockville, MD, USA, TR30022) according to manufacturer’s protocol. HEK293T cells were grown at 37 °C with 5% CO_2_ in Dulbecco’s modified Eagle’s medium (DMEM, Gibco, 12800017) supplemented with 10% FBS (Gibco, 10437028) and 1% (Pen-strep, Gibco, DW101203-031-1B). For Pacer loss of function experiments, MSC were seeded at a concentration of 2 × 10^5^ cells/well in six-well plates and transfected 24 h later with ON-TARGETplus smart-pool siRNAs targeting mouse Pacer (siPacer) and ON-TARGETplus non-targeting siRNAs as a control (siCtrl) (both from Dharmacon) using Dharmafect Transfection Reagents (Dharmacon, T-2001-01). Briefly, 4 μL Dharmafect was used to obtain a final concentration of 30 nM siRNA/well. After 48 h of transfection, the cells were co-cultured with splenocyte (obtained from C57BL/6 mice) or used in cell biology experiments. For Pacer gain-of-function experiments, lentiviral particles were produced in HEK 293T cells. Briefly, HEK293T cells were seeded at a concentration of 2.5 × 10^6^ in a 10 cm dish in 10 mL complete DMEM growth media (without antibiotic) and incubated overnight. Then, the cells were transfected with 5 µg of either empty vector or human Pacer Flag-tagged plus 6 µg of packaging plasmids from Lenti-ORF clones kit (Origene, TR30022). The medium was replaced 12 h post-transfection. The viral supernatant was collected at 24 h and 48 h and filtered through a 0.45 μm filter to remove cellular debris. High titer lentiviral stocks were produced (10^6^–10^7^ TU/mL). Murine MSC were transduced at passage 9 with lentivirus according to Lenti-ORF clones (Origene, TR30022), following the manufacturer’s instruction, generating MSC transduced with empty vector (EV-MSC) or with human Flag-tagged Pacer (hPacer-MSC). Both constructs carry a puromycin resistance, hence MSC resistant to puromycin (10 μg/mL) were selected over 3 passages and used in splenocyte co-culture assays and in vivo experiments up to passage 18. 

### 2.3. Autophagy Assays

To induce autophagy in MSC, cells were treated with rapamycin (200 nM, Enzo Life Sciences, Farmingdale, NY, USA, BML-A275), EBSS (Sigma, E2888) for 4 h and human TNFα (hTNFα) (10 µg/mL) for 2 and 4 h. For autophagy flux experiments, MSC cells were transfected with vectors encoding human Pacer-Flag or empty vector (OriGene) using TransIT-x2 dynamic delivery system (Mirus Bio, Madison, WI, USA, MR.MIR6000) according to manufacturer´s instructions or siRNA oligos targeting mouse Pacer or scrambled siRNAs as a control (Dharmacon) transfected with Dharmafect Transfection Reagents (Dharmacon, T-2001-01) using the manufacturer´s protocol. To induce autophagy, cells were treated with hTNFα (10 µg/mL) (Sigma, SRP3177) for 30 min, 2 h or 4 h. To inhibit autophagosome-lysosome fusion, cells were pretreated with Bafilomycin A1 for 30 min (0.5 µM, Sigma, B1793-10UG) and then stimulated with hTNFα at the same times described above. Autophagy flux was calculated as described in Ref. [37]: LC3II flux per sample equals LC3 II densitometric values (after normalization to β-Actin) of lysosomal inhibitor-treated samples minus lysosomal inhibitors untreated controls. This was performed for each independent experiment, and the 3 N were graphically represented. The same was performed for p62 flux.

### 2.4. Real-Time PCR 

Total RNA was extracted with TRIzol (Life Technologies, 10296028) and reverse transcribed with the First-strand cDNA synthesis kit (Thermo Fisher Scientific, Waltham, MA, USA, K1621). mRNAs levels were determined by real-time PCR using SYBR Green (Kappa Biosystem, Wilmington, MA, USA, KK4602) and normalized to mRNA levels of GAPDH or β-Actin. The primer sequences were as follows: mouse GAPDH, sense 5-TGTGATGGGTGTGAACCACGAGAA-3′ and antisense 5′-GAGCCCTTCCACAATGCCAAAGTT-3′; β-Actin, sense 5′-AAGATCATTGCTCCTCCTGA-3′ and antisense 5′-TACTCCTGCTTGCTGATCCA-3′; mouse Pacer, sense 5′–TTCACCCACCAATCAAGAGGGACA-3′ and antisense 5′-ACAAGACTCTGCAGATGAGTGGCA-3′; mouse PTGS2, sense 5-GAAGTCTTTGGTCTGGTGCCT-3′ and antisense 5′-TGCTCCTGCTTGAGTATGTCG-3′; human-mouse Pacer, sense 5-ACACTGACCATCCTCCTTGC-3′ and antisense 5-GTTGTCTCTGCCAGGGAGTC-3′; IL-6, sense 5-TGGTACTCCAGAAGACCAGAGG-3′ and antisense 5′-AACGATGATGCAGCACTTGCAGA-3′; Tgfb1, sense 5-CACTGATACGCCTGAGTG3′ and antisense 5′-GTGAGCGCTGAATCGAAA-3′. PCR conditions were: 1 cycle at 95 °C for 5 min, followed by 35 cycles at 95 °C for 30 s, 55 °C for 30 s and 72 °C for 2 min. The final extension step was carried out at 72 °C for 10 min. The PCR products were analyzed on 1% agarose gel. All primers were synthesized by Integrated DNA Technologies (San Diego, CA, USA).

### 2.5. Immunoblot 

MSC were lysed in RIPA buffer (NaCl 1 M; Nonidet P-40 1%; Sodium deoxycholate 0.5%; SDS 0.1%; Tris 50 mM, pH 7.4) containing the protease inhibitor (Roche, Basel, Switzerland, 88666) and phosphatase inhibitor (Roche, 4906837001), and sonicated for 15 s. The protein concentration was determined by BCA protein assay (Thermo Scientific, Waltham, MA, USA, 23227) according to manufacturer’s recommendations. Protein samples were used at 40 µg and heated to 95 °C for 5 min and run in SDS-polyacrylamide gel. Proteins were electroblotted onto polyvinylidene fluoride (PVDF, Thermo Scientific, 88518) membranes and blocked for 30 min in 5% bovine serum albumin (BSA, Merk, Darmstadt, Germany, 810037) in Tris-Buffer Saline (TBS, pH 7.6) at room temperature. PVDF membranes were incubated with primary antibodies overnight at 4 °C or 2 h at room temperature, then extensively washed with Tris-Buffer Saline containing 0.01% Tween (TBST), incubated with HRP-conjugated secondary antibody (The Jackson Laboratory, Bar Harbor, ME, USA) for 1 h at room temperature. Chemiluminescent reagent (Pierce ECL, Thermo Scientific) 32106 was used according to manufacturer´s instructions to visualize the detected proteins. The primary antibodies used were mouse anti-human-Pacer (1:1000) (Novus, Littleton, CO, USA, B01P, NBH00080183-B01P), mouse anti-PTGS2 (Cox2) (1:2000) (Beckton Dickinson, 610203), mouse anti-mouse Pacer (1:1000) [33], rabbit anti-LC3B (Cell Signaling Technology, 2575), 1:1000, mouse anti-SQSTM1/p62 (Abcam, Cambridge, UK, ab56416), 1:10,000, mouse anti-Flag (Sigma, F1804), 1:1000, rabbit anti-Beclin1 (Santa Cruz Biotechnology, Dallas, TX, USA, sc-11,427), 1:5000, and rabbit anti-GAPDH (Cell Signaling Technology, 2118s), or rabbit anti-β-Actin (Cell Signaling Technology, 4967) were used as loading controls, 1:2000 or 1:5000, respectively. Secondary HRP-conjugated anti-rabbit (Jackson Lab, 715035152), anti-mouse (Jackson Lab, 715035150) antibodies were employed at a 1:10,000 dilution.

### 2.6. T-Cell Proliferation Assay 

For MSC/splenocyte co-culture assays, 10,000 MSC were seeded per well into 48-well plates. Freshly isolated splenocytes (1 × 10^6^ cells/well) from female C57BL/6 (Jackson Lab) mice were labeled with 10 μM CellTraceTM Violet (CTV) (Invitrogen, Warrington, UK, C34571) according to the manufacturer’s instructions, stimulated with ConA (1 µg/mL) (Sigma, C5275) for T-cell activation and co-cultured with MSC in a ratio (1:10, MSC: splenocytes) in complete RPMI medium (Gibco, 11875-093) with 10% FBS (Gibco, 10437028), 100 U/mL penicillin and 100 μg/mL streptomycin (Gibco, DW101203-031-1B) at 37 °C and 5% CO_2_. After 5 days of co-culture, cells were washed and evaluated using flow cytometry. To assess T-cell proliferation, we used CTV. Each peak of the histograms corresponds to one cell division for CD3+ lymphocytes. The proliferation index was calculated as follows: the number of cells divided by the number of progenitors, as described by Roederer et al. [38]. Furthermore, cell survival was evaluated with SYTOX™ Green dye (Invitrogen, S7020). Flow cytometry was performed in a CytoFLEX (Beckman Coulter, Brea, CA, USA) and analyzed using the FlowJo 10.6 software.

### 2.7. DSS-induced Colitis Mouse Model

To model acute colitis, 2.5% dextran sulfate sodium (DSS) (Millipore, Burlington, MA, USA, MP-0216011080) was dissolved in drinking water and was orally administered to female C57BL/6 mice (14–16-weeks old) for 7 days. Murine MSC were intraperitoneally (i.p.) administered into mice on day 3. The mice were divided into four groups: (i) healthy controls not treated with DSS, (ii) mice treated with DSS, (iii) mice treated with DSS and i.p. injected with EV-MSC, and (iv) mice treated with DSS and i.p. injected with hPacer-MSC. A total of 1 × 10^6^ MSC in a volume of 100 μL phosphate buffer saline (PBS) (Gibco, 10010049) was used for i.p. injection. Weight loss, stool consistency/diarrhea and the presence of rectal bleeding were daily assessed. The Disease Activity Index (DAI) was calculated as previously described [39]. Briefly, the sum of scores for (i) weight loss (0–4), (ii) stool consistency/diarrhea (0–4) and (iii) rectal bleeding (0–4) was calculated. The DAI is classified as 0 (healthy mice) to 12 (severe colitis). Mice were sacrificed at day 14, and the colon length was measured previous to processing colon tissue for biochemical and histological analyses. 

### 2.8. Histological Evaluation and Immunohistochemistry (IHC)

The Swiss-roll technique was used to prepare tissues and perform the histological analyses of the mouse intestine. Colon samples were fixed in 4% paraformaldehyde; 5 μm-thick sections were stained with hematoxylin and eosin (H&E). Colonic inflammation was graded by a histopathological scoring system. The scores were determined as follows: No evidence of inflammation—Score 2; Low leukocyte infiltration (<10% of section), no structural damage—Score 4; moderate leukocyte infiltration restricted to the mucosal layer (10–25% of section) crypt, elongation, partial loss of goblet cells, bowel wall thickening, no ulcerations—Score 6; severe leukocyte infiltration beyond the mucosal layer (25–50% of section), crypt elongation, bowel wall thickening, superficial ulcerations—Score 8; transmural leukocyte infiltration seen in >50% of section, distorted crypts, marked loss of goblet cells, bowel wall thickening, extensive ulcerations—Score 10 [40]. The histopathological score was normalized by total colon area.

### 2.9. Statistical Analysis

Statistical analysis was performed with GraphPad Prism V7 software. Statistically significant values were determined using one-way ANOVA, two-way ANOVA or Student’s *t*-test. 

## 3. Results

### 3.1. The Autophagy Enhancer Pacer Is Upregulated in MSC in Response to the Pro-Inflammatory Cytokine TNFα 

We and others have recently described the novel autophagy protein Pacer to be part of the autophagic machinery required for responding to a higher demand for autophagic capacity in the context of neurodegenerative and liver disease [33,34]. Autophagy previously has been shown to be involved in the response of MSC to the inflammatory environment they face in various disease contexts, including experimental autoimmune encephalomyelitis (EAE), Alzheimer, SLE, among others [41,42,43]. To determine whether Pacer could be involved in the response of MSC to an inflammatory environment during colitis, we first assessed the levels of Pacer and other autophagy markers by Western blot in the presence of TNFα, since this cytokine plays an important role in the pathogenesis of colitis [44]. We found that the treatment with TNFα led to a significant upregulation of the protein levels of Pacer, the autophagy core component Beclin1 and the autophagy marker LC3II, as well as a significant downregulation of the autophagy substrate p62, similar to treatments with EBSS (starvation) or rapamycin, two conditions generally used to induce autophagy (Figure 1A,B). These results suggest an activation of the autophagy pathway in MSC while encountering a pro-inflammatory environment mainly defined by TNFα, such as the one found during IBD. Since Pacer has been reported to promote autophagic activity [33,34], we determined whether the presence of Pacer is related to the autophagic response to TNFα by performing autophagic flux assays under TNFα treatment for 0.5, 2 or 4 h, while also inhibiting lysosomal degradation with lysosomal inhibitors (Figure 1C). To deplete Pacer levels in MSC, we employed siRNA targeting Pacer mRNA (siPacer). As a control, scrambled siRNA (siCtrl) was used. Interestingly, we found that under TNFα stimulation, the levels of endogenous Pacer were notably increased under lysosomal inhibition, indicating its degradation by the lysosome (Figure 1C). On the other hand, we found that a decrease in Pacer expression by a knockdown in TNFα-stimulated MSC resulted in a decrease in LC3II levels under lysosomal inhibition, however, with no significant changes in p62 levels (Figure 1C,D), indicating impaired autophagy flux under these conditions. These results are in line with the reported role of Pacer as an autophagy enhancer [33,34]. Conversely, to investigate whether an increase in Pacer levels could improve autophagic capacity under TNFα stimulation, we overexpressed human Flag-tagged Pacer (hPacer) in MSC and determined the autophagic flux compared to an empty vector as a control (EV) (Figure 1E). We found that increased Pacer expression enhanced the autophagy activity of MSC, as indicated by an increase in LC3II under lysosomal inhibition, however, with no significant changes in p62 levels (Figure 1E,F). Together these results suggest that a pro-inflammatory environment results in the activation of autophagy in MSC and that this increase in autophagic capacity can be modulated by alterations in the levels of Pacer. 

### 3.2. Pacer Is Required for the Immunosuppressive Function of MSC

To investigate the response of MSC to TNFα, we determined the mRNA levels of several pro- or anti-inflammatory molecules, including, *IL6*, *TGF1β* and prostaglandin endoperoxide synthase 2 (*PTGS2*, also referred to as *COX2*) (Figure 2A). We found a significant increase in the mRNA levels of *IL6*, *TGF1β* and *PTGS2* in MSC in response to TNFα treatment (Figure 2A). PTGS2 is responsible for the synthesis of to the potent immunosuppressor Prostaglandin E2 [45], and a link between Beclin1 and PTGS2 has been reported previously [41]. Since PTGS2 is an important immune modulator in MSC, we confirmed its expression by determining its protein levels by Western blot (Figure 2B). We found increased levels of this enzyme at 2 h and 4 h of TNFα treatment (Figure 2B). Interestingly, rapamycin treatment but not starvation conditions (EBSS) also induced increased mRNA levels of *IL6*, TGF1β and *PTGS2*, as well as PTGS2 protein levels (Figure 2A,B). These results suggest that in MSC, the autophagy pathway is in part regulated by the levels of TNFα found in the tissue environment and that the autophagic capacity of MSC may influence their secretion of cytokines, hence their immunomodulatory potential. 

We hypothesized that the modulation of Pacer may play a role in the immunomodulatory functions of MSC, hence we determined the protein levels of PTGS2 under Pacer loss and gain of function (Figure 2C,D). We found that the depletion of Pacer in MSC resulted in diminished PTGS2 levels and a lack of PTGS2 upregulation upon TNFα treatment (Figure 2C), while the overexpression of Pacer resulted in significantly increased PTGS2 levels under non-treated conditions, which were maintained under TNFα treatment (Figure 2D). Hence, these results, together with our previous results, suggest that the immunomodulatory response of MSC to a TNFα pro-inflammatory environment may be dependent on Pacer expression levels. It has been previously reported that TNFα can enhance the immunosuppressive function of MSC [22]. Thus, we confirmed this observation by performing a splenocyte proliferation assay, where we co-cultured MSC stimulated with or without TNFα together with mouse splenocytes (Appendix A). To study the potential role of Pacer in regulating the immunosuppressive capacity of MSC, we depleted MSC of Pacer expression and then examined their effects on T-cell proliferation. By using RNA interference targeting Pacer mRNA (siPacer) and the appropriate nonsense control (siCtrl), we achieved a knockdown of Pacer mRNA and protein levels of approximately 50% (Figure 3A,B). Next, we assessed the immunosuppressive function of these cells. MSC with diminished Pacer levels suppressed the proliferation of activated T cells less efficiently than the MSC treated with siCtrl or untreated MSC (Figure 3C). To determine whether increased expression of Pacer in MSC would have the inverse effect, we generated MSC that overexpress human Flag-tagged Pacer (hPacer) using lentiviral transduction (Figure 3D,E) and then examined their effects on T-cell proliferation. MSC transduced with lentivirus carrying a construct for hPacer expression (hPacer-MSC) suppressed the proliferation of activated T cells more efficiently than MSC transduced with lentivirus carrying an empty vector (EV-MSC) (Figure 3F). Together, these results suggest that Pacer is required for the immunosuppressive capacity of MSC and that increasing Pacer levels may enhance this capacity. 

### 3.3. Increasing Pacer Levels in MSC Improves Their Therapeutic Effect in a Mouse Model of DSS-Induced Colitis

Since increased Pacer levels enhanced the immunosuppressive capacity of MSC in vitro, we tested whether hPacer-MSC have a comparable effect in vivo. To explore this potential effect, we used the DSS-induced acute inflammatory colitis mouse model. Feeding mice for several days with DSS polymers in the drinking water induces an acute colitis characterized by weight loss, bloody diarrhea, rectal bleeding, ulcerations and infiltrations with granulocytes. In our protocol, we used DSS at a concentration of 2.5% (*w*/*v*) in the drinking water for 7 days, which induces strong colitis with very low mortality rates [12]. EV-MSC or hPacer-MSC were intraperitoneally injected into the mice on day 3 of the DSS treatment (Appendix A). The overexpression of hPacer significantly improved the therapeutic effects of MSC toward intestinal injury, which was assessed by determining the disease activity index (Figure 4A), which included weight recovery (Appendix A). At day 14, the large intestine and cecum were extracted from the peritoneal cavity. The colon length was measured to assess general organ damage. The colon of DSS treated mice was significantly shorter than the colon of healthy control mice, whereas mice treated with DSS and injected with MSC overexpressing hPacer displayed a similar colon length as healthy control mice, which was also improved compared to mice treated with DSS and EV-MSC (Figure 4B,C). Histological analysis of hematoxylin/eosin-stained colon sections showed that the administration of hPacer-MSC was able to regenerate the damaged intestinal epithelium more efficiently than EV-MSC (Figure 4D). Furthermore, a lower level of inflammation with scattered infiltrating mononuclear cells (1–2 foci) was observed with hPacer-MSC treatment compared to EV-MSC treatment (Figure 4D,E). These results show that increased levels of Pacer improve the therapeutic capability of MSC to alleviate DSS-induced inflammatory colon injury.

## 4. Discussion

Several recent in vitro and preclinical studies have demonstrated the therapeutic potential of MSC while used as a treatment for diverse pathological conditions, e.g., diabetes, neurodegenerative diseases, inflammatory or autoimmune disorders, among others [8,46,47]. Initially, the therapeutic capacity of MSC had been attributed to their ability to differentiate into various cell lineages, hence replacing damaged cells [48]. Nonetheless, controversy exists around their competence to differentiate in vivo, as well as their potency to survive and engraft in the targeted tissue post-transplantation [7,49,50]. Hence, it was suggested that, additionally, MSC could create a healing environment through trophic actions upon local tissue cells, as well as immune cells recruited to the injury site. This process includes cell-to-cell contact, the secretion of growth factors, cytokines and extracellular vesicles [51]. Specifically, MSC display immunosuppressive properties, such as (i) inhibiting the activation and proliferation of B and T cells, (ii) impeding immune effector cell maturation, (iii) promoting regulatory immune cell expansion and (iv) constraining the inflammatory response by reducing the secretion of pro-inflammatory chemokines and cytokines (IL1b, IL2, TNFα, IFNɣ) and (v) promoting the release of anti-inflammatory factors (IL10, TGF1β) (reviewed in Ref [51]). However, the molecular mechanism associated with the immunomodulatory function of MSC remained poorly understood. 

Recent mounting evidence suggests that autophagy could play a central role in the ability of MSC to modulate the host immune response. For instance, autophagy is activated in MSC in an inflammatory environment [41,52], and this increase in autophagic capacity was found to improve their immunosuppressive and therapeutic functions [53,54]. Furthermore, Beclin1, a core component of the autophagy machinery, has been shown to be involved in the therapeutic properties of MSC in a murine encephalomyelitis model [41]. In this study, we investigated the potential role of the novel autophagy enhancer Pacer in the immunomodulatory properties of MSC. We found Pacer, Beclin1 and LC3II levels to be upregulated, while p62 levels were downregulated in MSC upon stimulation with the pro-inflammatory cytokine TNFα, one of the main cytokines present in the inflammatory environment of IBD, indicating an increase in the autophagy capacity of MSC under these conditions. Furthermore, Pacer loss- and gain-of-function experiments in combination with autophagy flux assays suggested that Pacer is a possible regulator/modulator of the autophagy machinery under TNFα pro-inflammatory condition, since its depletion in MSC resulted in impaired autophagy flux, while the augmentation of its levels resulted in enhanced autophagy flux. Additionally, a TNFα-mediated pro-inflammatory environment also upregulated the expression levels of immunomodulatory factors in MSC, such as *IL6*, *TGF1β* and PTGS2. Interestingly, we found PTGS2 levels to be highly dependent on the expression levels of *Pacer*, which suggests the Pacer may not only act as a regulator of autophagic capacity in MSC, but it could also be a modulator of the secretion of immunomodulatory factors of MSC, such as prostaglandins. The finding that alterations in Pacer levels can modulate the immunosuppressive capability of MSC further supports this conclusion. Our results show that Pacer loss of function led to impaired MSC control over splenocyte proliferation, while Pacer gain of function enhanced the expected immunosuppressive function of MSC to control splenocyte proliferation. Strikingly, these enhancing effects of increased *Pacer* expression in MSC were directly translatable to an acute in vivo mouse model of DSS-induced colitis, where MSC were used as a therapy. The increased expression of *Pacer* in MSC enhanced the ability of MSC to ameliorate the symptoms of DSS-induced colitis in mice. Our results are further supported by previous findings [31] that autophagy regulates the therapeutic potential of MSC and suggest that the augmentation of autophagic capacity in MSC by increasing Pacer levels may have therapeutic implications for IBD. Moreover, the modulation of MSC autophagy has been proposed as a possible strategy to favor MSC-induced T-cell polarization toward regulatory cells [55]. Thus, MSCs derived from adipose tissue pre-treated with rapamycin were more effective than untreated MSC in suppressing the in vitro expansion of T helper 17 cells and in reducing the mortality and clinical severity of acute graft versus host disease induced in mice [55]. In vivo benefits are associated with a reduction in T helper 17 cells and an increase in regulatory T cells. The potentiation of immunoregulatory MSC function was correlated with the activation of the autophagic machinery, since increased mRNA expression of some autophagy genes, such as *ATG5* and *LC3*, increased protein expression of Beclin1, ATG5, ATG7 and LC3-II, and concomitant suppression of the expression of *MTOR* and MTOR components (*RICTOR* and *RPTOR*) was found [55]. We demonstrate that the enhancement of autophagy in MSC by increasing Pacer levels improves the immunosuppressive effects of MSC toward T-cell proliferation, resulting in augmented therapeutic efficacy in vivo. These findings highlight the importance of autophagy in regulating the immunomodulatory function of MSC in the inflammatory microenvironment and suggest Pacer as a possible mechanistic link. 

In summary, our findings demonstrate a critical role of Pacer in the dual regulation of the inflammatory environment on the immunosuppressive function of MSC; while pro-inflammatory cytokines empower MSC to suppress immune responses, these cytokines increase Pacer expression, which in turn enhances autophagic capacity, improving the immunosuppressive function of MSC. Therefore, the modulation of Pacer in MSC may provide a novel strategy to improve MSC-based immunotherapy. Given its potent effects on reducing T-cell responses, it will be valuable to investigate whether the overexpression of Pacer in MSC may augment their therapeutic efficacy in other inflammatory-related diseases. 

## 5. Patents

Parts of the results presented in this study have been submitted for patent application.

## Figures and Tables

**Figure 1 cells-11-01503-f001:**
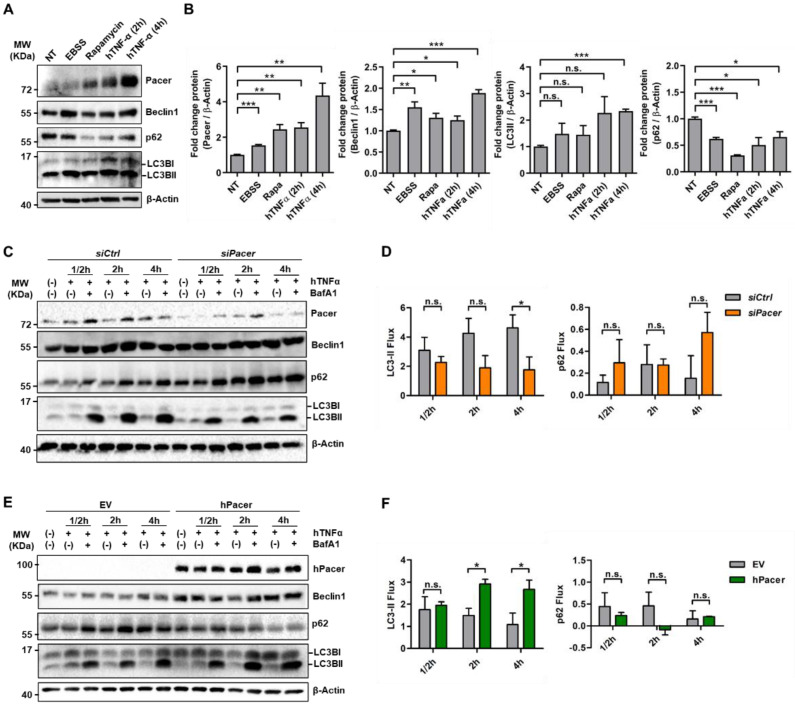
Pacer gain and loss of function affects autophagy flux in MSC treated with hTNFα. (**A**) MSC were treated with EBSS (4 h), Rapamycin (4 h) (200 nM) and hTNFα (10 µg/mL for 2 h or 4 h). Non-treated control cells are designated as NT. Western blot analysis showing the levels of mouse Pacer, Beclin1, SQSTM1/p62 and LC3B. β-Actin was used as a loading control. A representative of three independent experiments is shown. (**B**) Densitometric quantifications of Western blot analysis shown in (**A**) mouse Pacer, Beclin1, LC3BII and p62 normalized to β-Actin levels (*n* = 3). Student’s *t*-test was performed. Mean and SEM are shown: *, *p* ≤ 0.05; **, *p* ≤ 0.01; ***, *p* ≤ 0.001; n.s., non-significant. (**C**) Autophagy flux assay in MSC under Pacer knockdown (siPacer), compared to cells treated with scrambled siRNA (siCtrl). To induce autophagy, cells were treated with hTNFα (10 µg/mL) for 0.5, 2 and 4 h, while under treatment or not with Bafilomycin A1 (BafA1) (0.5 µM), to block lysosomal degradation. Pacer, Beclin1, p62 and LC3-II levels were determined by Western blot (*n* = 3). β-Actin serves as a loading control. A representative of three independent experiments is shown. (**D**) Densitometric quantifications of LC3II and p62 levels were performed, and autophagy flux was determined (*n* = 3). Student’s *t*-test was performed. Mean and SEM are shown: *, *p* ≤ 0.05; n.s., non-significant. (**E**) Autophagy flux was determined under Pacer gain of function (hPacer) compared to an empty vector control (EV). To induce autophagy, cells were treated with hTNFα (10 µg/mL) for 1/2, 2 and 4 h, while under treatment or not with Bafilomycin A1 (BafA1) (0.5 µM), to block lysosomal degradation. Pacer, Beclin1, p62 and LC3II levels were determined (*n* = 3). β-Actin serves as a loading control. A representative of three independent experiments is shown. (**F**) Densitometric quantifications of LC3II and p62 levels were performed, and autophagy flux was determined (*n* = 3). Student’s *t*-test was performed. Mean and SEM are shown: *, *p* ≤ 0.05; n.s., non-significant.

**Figure 2 cells-11-01503-f002:**
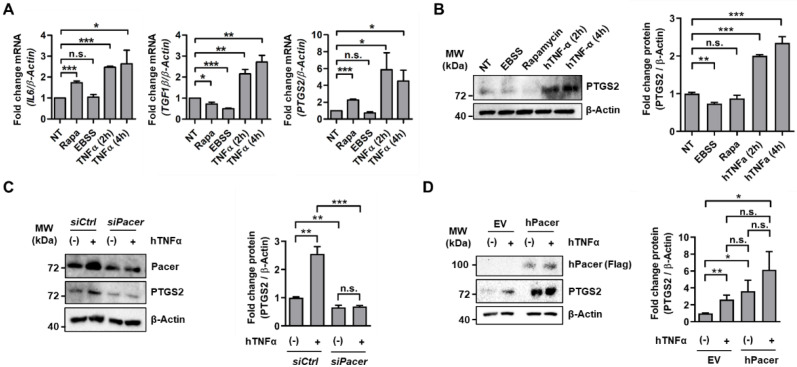
Pacer modulation affects PTGS2 levels in MSC. (**A**,**B**) MSC were treated with EBSS (4 h), Rapamycin (200 nM, 4 h) and hTNFα (10 µg/mL, 2 h or 4 h). Non-treated control cells are designated as NT. (**A**) mRNA levels of IL6, TGF1β and PTGS2 were determined by quantitative PCR (*n* = 3). *β -Actin* mRNA levels were used as a reference. Student’s *t*-test was performed. Mean and SEM are shown: *, *p* ≤ 0.05; **, *p* ≤ 0.01; ***, *p* ≤ 0.001; n.s., non-significant. (**B**) PTGS2 protein levels were determined by Western blot analysis (*n* = 3). β-Actin serves as a loading control. Densitometric quantifications were performed of three independent experiments. Student’s *t*-test was performed. Mean and SEM are shown: **, *p* ≤ 0.01; ***, *p* ≤ 0.001; n.s., non-significant. (**C**) PTGS2 protein levels under Pacer knockdown (siPacer) were determined by Western blot. Cells were treated with hTNFα (10 µg/mL) for 4 h. As a mock control, a scrambled siRNA (siCtrl) oligo was used. Mouse Pacer and PTGS2 levels of three independent experiments were detected (*n* = 3). β-Actin serves as a loading control. Densitometric quantification of PTGS2 levels normalized to β-Actin levels is shown (*n* = 3). Student’s *t*-test was performed. Mean and SEM are shown: **, *p* ≤ 0.01; and ***, *p* ≤ 0.001; n.s., non-significant. (**D**) PTGS2 protein levels were determined by Western blot under human Pacer (hPacer) overexpression. Cells were treated with hTNFα (10 µg/mL) for 4 h. As a mock control, an empty vector (EV) construct was used. hPacer and PTGS2 were detected in five independent experiments (*n* = 5). β-Actin serves as a loading control. Densitometric quantifications of PTGS2 normalized to β-Actin levels is shown (*n* = 5). Statistical analyses were performed using Student’s *t*-test. Mean and SEM are shown: **, *p* ≤ 0.01; and ***, *p* ≤ 0.001; n.s., non-significant.

**Figure 3 cells-11-01503-f003:**
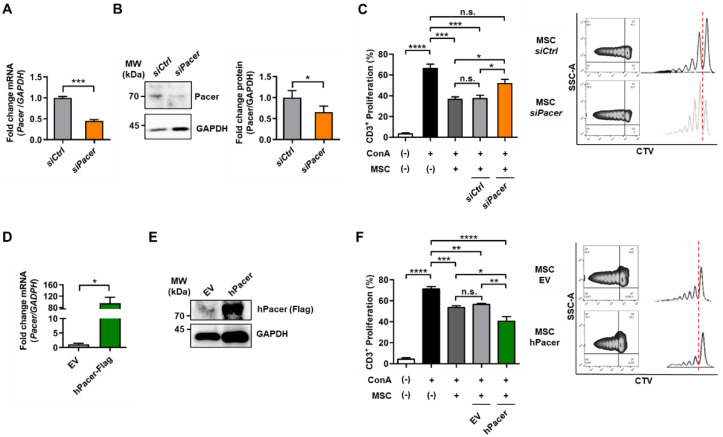
Increased levels of Pacer in MSC enhance their immunosuppressive capabilities upon T cells. (**A**–**C**) Depletion of endogenous Pacer levels in MSC using siRNA oligos. (**A**) mRNA levels of *Pacer* in MSC treated with si*Pacer* or si*Ctrl* oligos analyzed by quantitative PCR (*n* = 4). *GAPDH* mRNA levels are used as a reference. (**B**) Western blot and quantitative analysis of Pacer levels in MSC transiently transfected with si*Pacer* or si*Ctrl* oligos (*n* = 4). GADPH protein levels are used as a loading control. (**C**) Co-culture assay of CD3-positive (CD3+) splenocytes with MSC transiently transfected with si*Pacer* or si*Ctrl* oligos (*n* = 7). Splenocyte proliferation was induced with 1 μg/mL Concanavalin A (ConA). T-cell proliferation was evaluated using flow cytometry, gating SytoxGreen-negative staining (live CD3+ cells) and assessing cell trace violet (CTV) staining. (**D**–**F**) Augmented expression of *Pacer* in MSC transduced with lentiviral particles carrying human Flag-tagged Pacer (hPacer) or an empty control vector (EV). (**D**) mRNA levels of total *Pacer* in hPacer and EV MSC was determined by qPCR using primers recognizing mouse and human Pacer mRNA with comparable efficiency (*n* = 4). *GAPDH* mRNA levels are used as a reference. (**E**) Western blot analysis of Flag-tagged human Pacer expression in MSC. GAPDH protein levels were used as a loading control. (**F**) Co-culture assay of CD3-positive (CD3+) splenocytes with hPacer or EV MSC (*n* = 4). Splenocyte proliferation was induced with 1 μg/mL Concanavalin A (ConA). T-cell proliferation was evaluated using flow cytometry, gating SytoxGreen-negative staining (live CD3+ cells) and assessing cell trace violet (CTV) staining. In (**A**–**D**,**F**), Mean ± SEM are shown. In (**A**,**B**,**D**), Student’s *t*-test was performed (*n* = 4) and in (**C**,**F**), one-way ANOVA with Tukey post hoc test was performed for statistical analysis (*n* = 7). Only significant *p* values are shown. *p* values: *, *p* ≤ 0.05; **, *p* ≤ 0.01; ***, *p* ≤ 0.001; ****, *p* ≤ 0.0001.

**Figure 4 cells-11-01503-f004:**
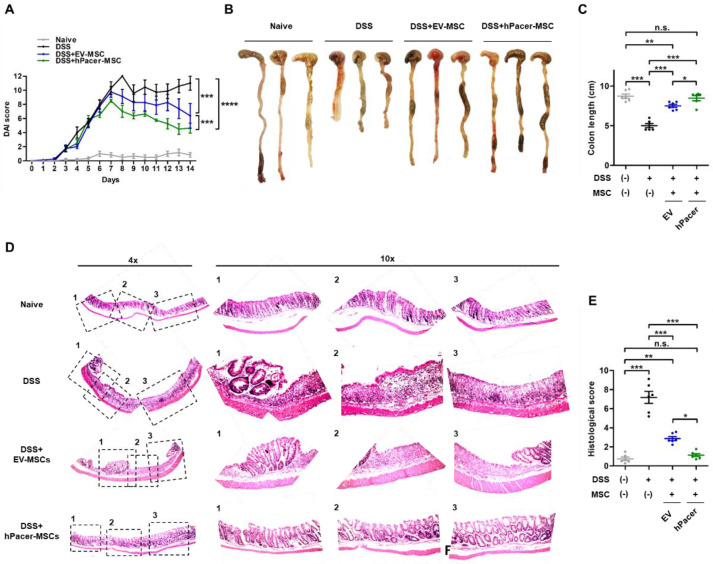
Upregulation of Pacer in MSC improves their therapeutic effects on DSS-induced inflammatory colon injury. (**A**) The disease activity index (DAI) scores of naïve (*n* = 6), DSS (*n* = 7), DSS+EV-MSC (*n* = 8) and DSS+hPacer-MSC (*n* = 8) were determined daily. The DAI was calculated from cumulative scores for body weight loss, stool consistency and presence of bleeding. (**B**,**C**) The colon length of each experimental group was measured, and the results were graphically displayed. (**D**,**E**) Histology of the colon was performed on day 14. (**D**,**E**) Representative images of the appearance of colon tissue are shown in (**D**) for naïve, DSS, DSS+EV-MSC and DSS+hPacer-MSC treated groups. (**E**) Images of each animal were analyzed, and histopathological scores were determined. (**A**,**C**,**E**) Mean ± SEM are shown. In (**A**), two-way ANOVA and in (**C**,**E**), one-way ANOVA with Tukey post hoc test were performed for statistical analysis. Only significant *p* values are shown. *p* values: *, *p* ≤ 0.05; **, *p* ≤ 0.01; ***, *p* ≤ 0.001; ****, *p* ≤ 0.0001.

## Data Availability

All data that lead to the conclusion of the study are shown in the manuscript or in Appendix A.

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
