# Peer review of "The Autophagy Protein Pacer Positively Regulates the Therapeutic Potential of Mesenchymal Stem Cells in a Mouse Model of DSS-Induced Colitis"

_cells, 2022, doi:10.3390/cells11091503_

Round 1
Reviewer 1 Report
In this manuscript, the authors evaluate the role of Pacer, a novel autophagy protein known to positively regulate autophagy, on the immunosuppressive capacity of MSC. The manuscript includes the role of Pacer in MSC upon TNFa stimulation, upon T-cell proliferation and in a DSS-induced colitis model in mice. The manuscript is well written and describes novel and important findings.
Some minor points need to be addressed before the manuscript can be accepted for publication:
On figure 1, the authors measure autophagic, LC3 flux or p62 flux, what does flux mean? Do they use BafA1 treated compared to non-BafA1 treated cells? Please define in the methods section how flux was calculated
For figure 1C,D, the authors state: we 282 found that a decrease in Pacer expression by knockdown in TNFα-stimulated MSC resulted in a decrease of LC3II and an increase in p62 levels under lysosomal inhibition 284 (Figure 1C and 1D). However, their numbers are non-significant. The authors should not claim increase or decreased levels with non-significant values. Same for figure 1 E,F.
Please check the reference format, in the text and in the reference list.
Author Response
In this manuscript, the authors evaluate the role of Pacer, a novel autophagy protein known to positively regulate autophagy, on the immunosuppressive capacity of MSC. The manuscript includes the role of Pacer in MSC upon TNFa stimulation, upon T-cell proliferation and in a DSS-induced colitis model in mice. The manuscript is well written and describes novel and important findings.
Some minor points need to be addressed before the manuscript can be accepted for publication:
- On figure 1, the authors measure autophagic, LC3 flux or p62 flux, what does flux mean? Do they use BafA1 treated compared to non-BafA1 treated cells? Please define in the methods section how flux was calculated
Response to reviewer: We thank the reviewer for her/his question and the opportunity to clarify. In our experimental setup we used TNFa treatment for 0.5, 2 or 4 hours to induce autophagic flux with or without simultaneous treatment with lysosomal inhibitor Bafilomycin. The autophagy flux was calculated according to Klionsky et al. Guidelines for the use and interpretation of assays for monitoring autophagy 3rd or 4th edition as follows: LC3II flux per sample equals LC3 II densitometric values (after Actin/loading control normalization) of lysosomal inhibitor-treated samples minus lysosomal inhibitors untreated controls. This was performed for each independent experiment and the 3 N were graphically represented. The same was done for p62 flux.
We included a new section in the methods to describe the calculation as follows:
“Autophagy flux was calculated as described in [37]: LC3II flux per sample equals LC3 II densitometric values (after normalization to β-Actin) of lysosomal inhibitor-treated samples minus lysosomal inhibitors untreated controls. This was performed for each independent experiment and the 3 N were graphically represented. The same was done for p62 flux.”
References
Klionsky et al. Guidelines for the use and interpretation of assays for monitoring autophagy (4th edition). Autophagy. 2021 Jan;17(1):1-382. doi: 10.1080/15548627.2020.1797280.
Klionsky et al. Guidelines for the use and interpretation of assays for monitoring autophagy (3rd edition). Autophagy. Autophagy. 2016;12(1):1-222. doi: 10.1080/15548627.2015.1100356.
- For figure 1C,D, the authors state: we 282 found that a decrease in Pacer expression by knockdown in TNFα-stimulated MSC resulted in a decrease of LC3II and an increase in p62 levels under lysosomal inhibition 284 (Figure 1C and 1D). However, their numbers are non-significant. The authors should not claim increase or decreased levels with non-significant values. Same for figure 1 E,F.
Response to reviewer: We agree with the reviewer’s comments regarding p62. We have modified the text according to the suggestions of the reviewer as follows:
“On the other hand, we found that a decrease in Pacer expression by knockdown in TNFα-stimulated MSC resulted in a decrease of LC3II levels under lysosomal inhibition however with no significant changes in p62 levels (Figure 1C and 1D), indicating impaired autophagy flux under these conditions. These results are in line with the reported role of Pacer as an autophagy enhancer 33, 34. Conversely, to investigate if an increase of Pacer levels could improve autophagic capacity under TNFα stimulation, we overexpressed human Flag-tagged Pacer (hPacer) in MSC and determined the autophagic flux compared to an empty vector as a control (EV) (Figure 1E). We found that increased Pacer expression enhanced the autophagy activity of MSC, as indicated by an increase in LC3II under lysosomal inhibition however with no significant changes in p62 levels (Figure 1E and 1F). Together these results suggest that a pro-inflammatory environment results in the activation of autophagy in MSC and that this increase in autophagic capacity can be modulated by alterations in the levels of Pacer.”
- Please check the reference format, in the text and in the reference list.
Response to reviewer: We thank the reviewer for her/his observation, we have corrected the reference formatting.
Reviewer 2 Report
The article presented by Cristian A. Bergmann and collaborates, entitled “The autophagy protein Pacer positively regulates the therapeutic potential of mesenchymal stem cells in a mouse model of DSS-induced colitis”, is an original article that aimed to investigate the role of Pacer- a novel autophagy-related protein associated with UVRAG as autophagy enhancer- in regulating the immunosuppressive capacity of MSC through its function in the autophagy pathway. The work is well designed, with good experiments and correct controls. The stated objectives are coherent and are answered throughout the work.
Major revision:
- In the Figure 1 C, D, E and F the authors analyze the function of Pacer as autophagy promotor in the presence of TNFalpha. The authors should specify in the graph of figure 1D/F if the quantification is carried out with the lysosomal inhibitor or without it. They should also quantify the rest of the parameters (Beclin...) In addition, the authors comment in the figure caption that they have an n of 3 and in the supplementary figures they only show one n.
- Is more correct the expression LC3II/I relation than LC3II in graphs
- Figure 4. The ability of these injected MSCs to migrate to the intestinal mucosa has been shown previously?
Minor revision
- The sentence (line 89) “Gao et al. demonstrated that the autophagy inhibitor 3‐methyl adenine (3‐MA) weakens the immunosuppressive function of MSC, whereas autophagy inducer rapamycin enhances this function mediated by TGF1β secretion 31. Interestingly, MSC treated with rapamycin aggregate while facilitating full-layer cutaneous wound healing and regeneration mediated by VEGF (Vascular Endothelial Growth Factor) secretion 32” needs to be rewritten as it is not well understood, in line?….while?
- The authors should describe the UVRAG acronym (line 97). In this line, the authors should improve the description of the role of Pacer in autophagy. The information is too brief in the introduction and later Pacer is the center of the work.
- Grammatical errors: 24 h not 24h (line 137); 12h (lines 148), 4h (line 159), etc; in vitro/in vivo on italic (lines 155, 339. 480. 517, 527, etc); Tgf1β (line 509)
- Figure 2A and 2B mRNA and protein same order of samples. In mRNA the NT group without error bar and WB has an error bar? I think it's the other way around
- Figure 3A and 3B/D. The Y axis is confusing. The authors should take more care of the forms in the graphs. It is more correct to put Pacer/GAPDH mRNA expression (Fold induction) or Pacer/GAPDH protein expression (Fold induction), also it does not give an error option. On the other hand, the authors must take into account that it is not appropriate to change housekeeping within the same job.
- References in text between []
- Duplicate numbers
Author Response
The article presented by Cristian A. Bergmann and collaborates, entitled “The autophagy protein Pacer positively regulates the therapeutic potential of mesenchymal stem cells in a mouse model of DSS-induced colitis”, is an original article that aimed to investigate the role of Pacer- a novel autophagy-related protein associated with UVRAG as autophagy enhancer- in regulating the immunosuppressive capacity of MSC through its function in the autophagy pathway. The work is well designed, with good experiments and correct controls. The stated objectives are coherent and are answered throughout the work.
Major revision:
- In the Figure 1 C, D, E and F the authors analyze the function of Pacer as autophagy promotor in the presence of TNFalpha. The authors should specify in the graph of figure 1D/F if the quantification is carried out with the lysosomal inhibitor or without it. They should also quantify the rest of the parameters (Beclin...) In addition, the authors comment in the figure caption that they have an n of 3 and in the supplementary figures they only show one n.
Response to reviewer: We thank the reviewer for her/his comments and the opportunity to clarify. In Figure 1 C and 1E we show the western blots of one of the 3 experiments performed, however for the graphical display in Figure 1 D and 1F of the autophagic flux we used the densiometric measurements of all 3 experiments. We also performed the analysis for Beclin1 as was suggested by the reviewer, however no changes were observed between TNF treatment and Ctrl. In general, in our understanding, for autophagic flux focus is put on the levels of LC3II and p62, hence we did not include the graphs for Beclin1 since it does not add more information than is provided by the Western blot image.
- Is more correct the expression LC3II/I relation than LC3II in graphs
Response to reviewer: Respectfully we disagree with the reviewer in this point. We determined the autophagic flux according to the recommendations outlined in Klionsky et al. Guidelines for the use and interpretation of assays for monitoring autophagy 2016 and 2021 published in Autophagy. Which does not recommend the use of the ratio of LC3II/I.
The autophagy flux was calculated according to the guidelines as follows: LC3II flux per sample equals LC3 II densitometric values (after Actin/loading control normalization) of lysosomal inhibitor-treated samples minus lysosomal inhibitors untreated controls. This was performed for each independent experiment and the 3 N were graphically represented. The same was done for p62 flux.
We included a new section in the methods to describe the calculation as follows:
“Autophagy flux was calculated as described in [37]: LC3II flux per sample equals LC3 II densitometric values (after normalization to β-Actin) of lysosomal inhibitor-treated samples minus lysosomal inhibitors untreated controls. This was performed for each independent experiment and the 3 N were graphically represented. The same was done for p62 flux.”
References
Klionsky et al. Guidelines for the use and interpretation of assays for monitoring autophagy (4th edition). Autophagy. 2021 Jan;17(1):1-382. doi: 10.1080/15548627.2020.1797280.
Klionsky et al. Guidelines for the use and interpretation of assays for monitoring autophagy (3rd edition). Autophagy. Autophagy. 2016;12(1):1-222. doi: 10.1080/15548627.2015.1100356.
- Figure 4. The ability of these injected MSCs to migrate to the intestinal mucosa has been shown previously?
Response to reviewer: We appreciate the interest and question of the reviewer and try to clarify as follows. MSC emerged as a new therapeutic opportunity for inflammatory diseases due to the capacity to improve and repair damaged tissues, however, few reports in the literature have put attention on clinical relevant injection routes. In Castelo-Branco et al. Technetium-99m-labeled MSCs were found to migrate towards the inflamed colon which was followed for 24 hours using a gamma camera. Furthermore, intraperitoneal injected MSC labeled with a lipophilic fluorochrome were mainly observed in the muscular layer and the submucosa and in the lamina propria of the mucosa, predominantly in the bottom of crypts in rats treated with trinitrobenzene sulfonic acid an established model for inducing colitis. Therefore, successful treatment of experimental colitis was found by using intraperitoneal administration and MSC migration to the inflamed colon. In Wang et al. three different MSC delivery routes were compared intraperitoneal, intravenous, and anal injections in the DSS-induced colitis mouse model. First, they demonstrated that MSC intraperitoneal injection has the highest survival rate of 87.5%, displayed less weight loss, quick weight gain, and complete absence of occult blood in DSS mice. Interestingly, DiR labeling of MSC showed a spread cell distribution of the injected MSC mainly in the large bowel. Therefore, to observe in more detail the MSC localization they use GFP+ MSC (collected from transgenic mice), these cells were found in the epithelium and mucosa of the inflamed colon 24h after MSC administration. Sala et al. found MSC aggregates in the peritoneal cavity of DSS mice, showing that these cell aggregates can secrete tumor necrosis factor-induced protein 6 (TSG-6) decreasing intestinal inflammation by immune system regulation.
References
Castelo-Branco MT, Soares ID, Lopes DV, Buongusto F, Martinusso CA, do Rosario A Jr, Souza SA, Gutfilen B, Fonseca LM, Elia C, Madi K, Schanaider A, Rossi MI, Souza HS. Intraperitoneal but not intravenous cryopreserved mesenchymal stromal cells home to the inflamed colon and ameliorate experimental colitis. PLoS One. 2012;7(3):e33360. doi: 10.1371/journal.pone.0033360.
Wang M, Liang C, Hu H, Zhou L, Xu B, Wang X, Han Y, Nie Y, Jia S, Liang J, Wu K. Intraperitoneal injection (IP), Intravenous injection (IV) or anal injection (AI)? Best way for mesenchymal stem cells transplantation for colitis. Sci Rep. 2016 Aug 4;6:30696. doi: 10.1038/srep30696.
Sala E, Genua M, Petti L, Anselmo A, Arena V, Cibella J, Zanotti L, D'Alessio S, Scaldaferri F, Luca G, Arato I, Calafiore R, Sgambato A, Rutella S, Locati M, Danese S, Vetrano S. Mesenchymal Stem Cells Reduce Colitis in Mice via Release of TSG6, Independently of Their Localization to the Intestine. Gastroenterology. 2015 Jul;149(1):163-176.e20. doi: 10.1053/j.gastro.2015.03.013.
Minor revision
- The sentence (line 89) “Gao et al. demonstrated that the autophagy inhibitor 3‐methyl adenine (3‐MA) weakens the immunosuppressive function of MSC, whereas autophagy inducer rapamycin enhances this function mediated by TGF1β secretion 31. Interestingly, MSC treated with rapamycin aggregate while facilitating full-layer cutaneous wound healing and regeneration mediated by VEGF (Vascular Endothelial Growth Factor) secretion 32” needs to be rewritten as it is not well understood, in line?….while?
- The authors should describe the UVRAG acronym (line 97). In this line, the authors should improve the description of the role of Pacer in autophagy. The information is too brief in the introduction and later Pacer is the center of the work.
- Grammatical errors: 24 h not 24h (line 137); 12h (lines 148), 4h (line 159), etc; in vitro/in vivo on italic (lines 155, 339. 480. 517, 527, etc); Tgf1β (line 509)
- Figure 2A and 2B mRNA and protein same order of samples. In mRNA the NT group without error bar and WB has an error bar? I think it's the other way around
- Figure 3A and 3B/D. The Y axis is confusing. The authors should take more care of the forms in the graphs. It is more correct to put Pacer/GAPDH mRNA expression (Fold induction) or Pacer/GAPDH protein expression (Fold induction), also it does not give an error option. On the other hand, the authors must take into account that it is not appropriate to change housekeeping within the same job.
- References in text between []
- Duplicate numbers
Response to reviewer: We thank the reviewer for her/his suggestions and observation and have corrected the manuscript accordingly. Besides correcting the grammatical errors and improved the labelling of the axis in the figures, we have included a new section in the Introduction to describe more previous findings about Pacer as follows:
“Pacer associates with Beclin1, and positively regulates autophagosome maturation by complex association with UVRAG (UV radiation resistance-associated gene protein) and stimulation of Vps34 kinase activity [33, 34, 36] . Pacer was shown to be modulated through phosphorylation by the mammalian target of rapamycin complex 1 (mTORC1) which prevents the interaction of Pacer with STX17 and the HOPS complex resulting in disrupted autophagosome maturation [35]. Dephosphorylation of Pacer promotes its acetylation in the RH domain which was shown to enhances the interaction with the above-mentioned proteins and promotes autophagosome maturation [35]. It has been suggested that Pacer has an important role in autophagy-mediated hepatic lipolysis to alleviate liver inflammation and injury [35]. Furthermore, Pacer has been shown to be involved in amyotrophic lateral sclerosis (ALS) pathogenesis where its loss of function led to increased ALS-associated protein aggregation and neuronal cell death [33]. A role for Pacer in MSC autophagic capacity or immunosuppressive function has not been investigated previously.”